# Chimeric Antigen Receptor (CAR) T Cells Releasing Soluble SLAMF6 Isoform 2 Gain Superior Anti-Cancer Cell Functionality in an Auto-Stimulatory Fashion

**DOI:** 10.3390/cells14120901

**Published:** 2025-06-14

**Authors:** Dennis Christoph Harrer, Tim Schlierkamp-Voosen, Markus Barden, Hong Pan, Maria Xydia, Wolfgang Herr, Jan Dörrie, Niels Schaft, Hinrich Abken

**Affiliations:** 1Department of Internal Medicine III—Hematology and Internal Oncology, University Hospital Regensburg, 93053 Regensburg, Germany; 2Leibniz Institute for Immunotherapy, Division Genetic Immunotherapy, University Regensburg, 93053 Regensburg, Germanyhong.pan@ukr.de (H.P.); hinrich.abken@ukr.de (H.A.); 3Bavarian Cancer Research Center (BZKF), 91054 Erlangen, Germany; jan.doerrie@uk-erlangen.de (J.D.); niels.schaft@uk-erlangen.de (N.S.); 4Department Dermatology, Universitätsklinikum Erlangen, Friedrich-Alexander-Universität Erlangen-Nürnberg, 91054 Erlangen, Germany; 5Comprehensive Cancer Center Erlangen European Metropolitan Area of Nuremberg (CCC ER-EMN), 91054 Erlangen, Germany; 6Deutsches Zentrum Immuntherapie (DZI), 91054 Erlangen, Germany

**Keywords:** CAR T cell, adoptive T cell therapy, checkpoint, stimulation, exhaustion

## Abstract

T cells equipped with chimeric antigen receptors (CARs) have evolved into an essential pillar of lymphoma therapy, reaching second-line treatment. In solid cancers, however, a dearth of lasting CAR T cell activation poses the major obstacle to achieving a substantial and durable anti-tumor response. To extend T cell cytotoxic capacities, we engineered CAR T cells to constitutively release an immunostimulatory variant of soluble SLAMF6. While wild-type SLAMF6 induces T cell exhaustion, CAR T cells with the soluble Δ17-65 SLAMF6 variant exhibited refined, CAR redirected functionality compared to canonical CAR T cells. CD28-ζ CAR T cells releasing soluble SLAMF6 increased IFN-γ secretion and augmented CD25 upregulation on CD4^+^ CAR T cells upon CAR engagement by pancreatic carcinoma and melanoma cells. Moreover, under conditions of repetitive antigen encounter, SLAMF6-secreting CAR T cells evinced superior cytotoxic capacity in the long term. Mechanistically, SLAMF6-secreting CAR T cells showed predominantly a central memory phenotype, a PD-1^-^ TIGIT^-^ double negative profile, and reduced expression of exhaustion-related transcription factors IRF-4 and TOX with augmented amplification and persistence capacities. Overall, CAR T cells engineered with the release isoform 2 SLAMF6 establish an auto-stimulatory loop with the potential to boost the cytolytic attack against solid tumors.

## 1. Introduction

In recent years, chimeric antigen receptor (CAR) T cells have morphed into an essential element of tumor immunotherapy, reaching the FDA and EMA approval for the treatment of certain B-cell lymphoid malignancies [1,2]. Currently, a growing body of clinical trials is exploring CAR T cells in the wide realm of solid tumors [3,4,5,6]. In striking contrast to the treatment of leukemia/lymphoma, remissions in solid tumor patients are rare so far, necessitating further preclinical refinement of currently available CAR T cell formats [7,8,9,10,11].

The lack of enduring capacities of CAR T cells to execute a cytotoxic anti-cancer cell response, which is a key hallmark of acquired T cell dysfunctionality, also denoted as “exhaustion”, is believed to be a vital obstacle to successful CAR T cell therapy of solid tumors [9,12,13]. In exhausted CAR T cells, the reduced killing capacity is accompanied by low cytokine production, entry into terminal differentiation, and upregulation of inhibitory receptors, such as PD-1 and TIGIT [14,15,16]. Various strategies to counteract exhaustion-related CAR T cell dysfunctionality have been explored during the last years, involving the knockdown of inhibitory receptors like PD-1, TIM3, LAG3, and TIGIT [16,17,18,19] and the repression of exhaustion-promoting transcription factors, like TOX1, IRF4, and BLIMP-1 [20,21,22]. Alternatively, transcription factors that preserve CAR T cell fitness, including FOXO1 and c-MYB, were overexpressed, or CAR T cells were supplied with cytokines that delay the onset of exhaustion, e.g., IL-7 or IL-15 [23,24,25,26,27]. Despite some success, the quest for additional strategies to improve CAR T cell functionality against solid tumors remains a high priority.

In this context, SLAMF6 (signaling lymphocytic activation molecule family-6) is raising particular interest since it was identified as a favorable target for augmenting the functionality of CD8^+^ T cells against melanoma cells, both in vitro and in vivo [28,29]. SLAMF6, which constitutes an immunomodulatory receptor comprising a variable and a constant chain with homotypic binding capacities, is expressed on T cells, B cells, and natural killer (NK) cells [28]. Previously, Eisenberg et al. employed a truncated soluble version of the SLAMF6 ectodomain to boost amplification, cytokine secretion, and cytolytic capacity of melanoma-specific CD8^+^ T cells [29]. Contrary to IL-2 supplementation, soluble SLAMF6 protected effector T cells from activation-induced cell death and polarized T cells towards a more central memory phenotype and functionality [29]. Notably, systemic infusion of soluble SLAMF6 into melanoma-bearing mice enhanced the persistence and performance of melanoma-specific T cells [29]. Three splice isoforms of SLAMF6 with different ectodomains have been described in T cells, with isoform 1 being the predominant variant expressed at the highest level, whereas isoform 2 lacks amino acids 17–65 (Δ17–65) of the variable region, and isoform 3 is devoid of the entire variable chain (ΔExon 2) [30]. Importantly, the SLAMF6 splice variants display different activities. The canonical SLAMF6 promotes T cell exhaustion, while the Δ17–65 isoform 2 sustains T cell functionality [30].

We intended to equip CAR T cells with the soluble immunostimulatory Δ17–65 isoform 2 of SLAMF6 to expedite their anti-cancer cell functionality in an auto-stimulatory loop. Technically, T cells were engineered with a 2nd generation CAR and a constitutive expression cassette for soluble SLAMF6 to obtain CAR T cells releasing soluble SLAMF6. Compared to conventional CAR T cells, SLAMF6-releasing CAR T cells displayed superior cytotoxicity and cytokine release against pancreatic adenocarcinoma and melanoma cells. Mechanistically, SLAMF6 CAR T cells retained a larger proportion in the central memory compartment and a higher percentage of CAR T cells double negative for PD-1 and TIGIT. This is the first study leveraging the potential of auto-stimulatory SLAMF6 release to boost CAR T cell functionality against solid tumors.

## 2. Materials and Methods

### 2.1. Cells and Reagents

Peripheral blood mononuclear cells (PBMCs) were procured by Lymphoprep^®^ centrifugation (Axis-Shield, Oslo, Norway) from healthy donors upon informed consent and approval by the institutional review board (21-2224-101 Regensburg). T cells were maintained in RPMI 1640 medium, 1% (*w*/*v*) GlutaMAX (Gibco, ThermoFisher, Waltham, MA, USA), 100 IU/mL penicillin, 100 µg/mL streptomycin (Pan-Biotech, Aidenbach, Germany), 2 mM HEPES (PAA, Palo Alto, CA, USA), and 10% (*v*/*v*) heat-inactivated fetal calf serum (FCS) (Pan-Biotech, Aidenbach, Germany). BxPC-3 cells (ATCC CRL-1420), 293T cells (ATCC CRL-3216 American Type Culture Collection, Manassas, VA, USA), and A375M were cultured in DMEM, 1% (*w*/*v*) GlutaMAX (Gibco, ThermoFisher), 100 IU/mL penicillin, 100 µg/mL streptomycin (Pan-Biotech), and 10% (*v*/*v*) heat-inactivated FCS (Sigma-Aldrich, St. Louis, MO, USA).

### 2.2. CAR T Cell Generation

Cryopreserved blood lymphocytes were thawed and stimulated on the same day with the anti-CD3 monoclonal antibody (mAb) OKT-3, the CD28 mAb 15E8, and IL-2 (1000 IU/mL). Recombinant IL-2 (200 IU/mL) was added on days 2, 3, and 4. Retroviral transduction was performed as previously detailed [21]. Before experimental use, CAR T cells were isolated via magnetic activated cell sorting (MACS) (Miltenyi Biotec, Bergisch Gladbach, Germany). The CEA-specific CAR BW431/26scFv-Fc-CD28-ζ-P2A-GFP expression cassette and the CSPG4-specific CAR (CSPG4_HL_ CD28-CD3ζ) were previously published [31,32]. The vectors encoding the CEA-specific CAR linked by a P2A-peptide with soluble SLAMF6 (extracellular part of NM_001184715.2) and the CSPG4-specific CAR linked by a P2A-peptide with either GFP or soluble SLAMF6 (extracellular part of NM_001184715.2) were synthesized by GenScript Biotech (Piscatawy, NJ, USA).

### 2.3. Flow Cytometry

Cells were labeled with antibodies at 4 °C for 15 min. For intracellular staining, cells were processed using the “Transcription Factor Buffer Set” (BD Biosciences, Franklin Lakes, NJ, USA) for 30 min at 4 °C. The viability dye eFluor 780 (ThermoFisher, Waltham, MA, USA) was employed for live/dead discrimination. Fluorescent-minus-one (FMO) controls were used to set gates. The goat F(ab′)2 anti-human IgG-PE antibody, which was used to detect the CAR, was purchased from SouthernBiotech, FITC-conjugated anti-CD8 (clone BW135/80), APC-conjugated anti-CD4 (clone VIT4), PE-conjugated anti-CD25 (clone 4E3), and APC-conjugated anti-Granzyme B (clone REA 226) were purchased from Miltenyi Biotec, BV421-conjugated anti-CD8 (clone RPA-T8), PerCP-Cy5.5-conjugated anti-TIGIT (clone A15153G), PE-conjugated anti-IRF-4 (clone IRF4.3E4), and BV421-conjugated anti-CD3 (clone OKT3) were acquired from Biolegend (San Diego, CA, USA), while BV421-conjugated anti-TIM3 (clone 7D3), BV421-conjugated anti-CD62L (clone DREG-56), BV605-conjugated anti-CD45RO (clone UCHL1), BV421-conjugated anti-Ki67 (clone Ki-67), BV421-conjugated anti-PD-1 (clone EH12.1), and PE-conjugated anti-SLAMF6 (clone hSF6.4.20) were obtained from BD Biosciences. The PE-conjugated anti-TOX (clone TXRX10) antibody was purchased from ThermoFisher. For analysis of the proliferative capacity, CAR T cells were incubated with 10 µM “Cell Proliferation Dye eFluor^®^ 450” (ThermoFisher) before stimulation. Immunofluorescence was determined using a BD FACSLyric (BD Biosciences). Data were analyzed using the FlowJo software version 10.7.1 Express 5 (BD Biosciences).

### 2.4. Cytokine Secretion

Target cells were seeded in 96-well round-bottom plates (1 × 10^5^ cells/well) overnight before adding CAR T cells (1 × 10^5^ cells/well). After 48 h of co-culture, IL-2 and IFN-γ in culture supernatants were determined by ELISA as previously described [33]. Detection of soluble SLAMF6 was performed after culturing CAR T cells for 48 h in medium and harvesting the supernatant for analysis using the Human NTB-A/SLAMF6 ELISA Kit (ThermoFisher).

### 2.5. Cytotoxicity Assay

CAR T cells (0.125-10 × 10^4^ cells/well) were co-incubated for 24 h in 96-well round-bottom plates with tumor cells (1 × 10^4^ cells/well) at the indicated effector to target ratios. The XTT-based colorimetric assay using the “Cell Proliferation Kit II” (Roche Diagnostics, Mannheim, Germany) was utilized to analyze specific cytotoxicity. The percentage of viable tumor cells in experimental wells was determined as follows: viability (%) = [OD (experimental wells − corresponding number of T cells)]/[OD (tumor cells without T cells − medium)] × 100. Cytotoxicity (%) was defined as 100 − viability (%).

### 2.6. Repetitive Stimulation Assay

GFP-expressing CEA^+^ BxPC-3 or GFP-expressing CSPG4^+^ A375M cells were seeded in 12-well plates (10^5^ cells per well). After 24 h, 10^5^ CAR T cells were added per well. After three days (Round 1, R1), the cells were harvested from the wells and resuspended in 1 mL T cell medium. A total of 100 μL was used for cell counting (live GFP^+^ tumor cells and live CD3^+^/CAR^+^ CAR T cells) via flow cytometry using counting beads (“CountBright”, ThermoFisher). The remaining 900 μL were added to a new 12-well plate with 10^5^ BxPC-3 cells or GFP-expressing CSPG4^+^ A375M cells for four days (round 2, R2). The procedure was repeated for round 3 (R3) or round 4 (R4), as indicated.

### 2.7. Statistical Analysis

Statistical analysis was performed with GraphPad Prism, Version 9 (GraphPad Software, San Diego, CA, USA). *p* values were determined by two-way ANOVA or Student’s t-test in conjunction with Welch’s correction as indicated; ‘ns’ indicates not significant, * *p* ≤ 0.05, ** *p* ≤ 0.01, and *** *p* ≤ 0.001.

## 3. Results

### 3.1. Design of CAR T Cells Releasing Soluble SLAMF6 Isoform 2

To generate CAR T cells releasing soluble SLAMF6, we engineered a retroviral expression cassette coding for both a CD28-ζ CAR targeting carcinoembryonic antigen (CEA) and the ectodomain of the immunostimulatory Δ17–65 isoform 2 of SLAMF6 (aCEA-28ζ-sSF6) (Figure 1A). The CAR with co-expressed GFP (aCEA-28ζ-GFP) in the same vector architecture served as a control [31]. Upon transduction, the CARs of both constructs aCEA-28ζ-sSF6 and aCEA-28ζ-GFP were expressed at similar levels on the T cell surface (Figure 1B). CAR T cells were isolated by magnetic cell sorting (MACS), yielding a purity of > 90% CAR^+^ T cells for further analyses (Figure 1B). The uniform expression of the endogenous membrane-anchored SLAMF6, which serves as a homotypic receptor for soluble SLAMF6, was confirmed by flow cytometry both on CD4^+^ and CD8^+^ CAR T cells (Figure 1C). Soluble SLAMF6 was constitutively secreted by aCEA-28ζ-sSF6 CAR T cells into the culture supernatant in the range of 700 pg/mL per 1 × 10^6^ cells/mL within 48 hrs as monitored by ELISA. No SLAMF6 secretion was detected in the supernatant of aCEA-28ζ-GFP CAR T cells (Figure 1D).

### 3.2. CAR T Cells Releasing Soluble SLAMF6 Isoform 2 Display Increased IFN-γ Secretion

We then investigated the impact of the secreted SLAMF6 isoform 2 on CAR T cell functionality. CD4^+^ aCEA-28ζ-sSF6 CAR T cells displayed a greater magnitude of CD25 upregulation upon co-culture with CEA^+^ BxPC-3 tumor cells in comparison to CD4^+^ aCEA-28ζ-GFP CAR T cells. No significant difference was observed for CD8^+^ CAR T cells (Figure 2A). Both aCEA-28ζ-sSF6 T cells and control aCEA-28ζ-GFP CAR T cells were equally capable of eliminating CEA^+^ BxPC-3 cells across different effector to target cell ratios without causing substantial cytotoxicity against CEA^-^ 293T cells (Figure 2B). Secretion of IFN-γ in response to BxPC-3 cells was significantly higher for CAR T cells with SLAMF6 release compared to control CAR T cells (Figure 2C). However, IL-2 secretion did not differ between SLAMF6-releasing and control CAR T cells (Figure 2C). Finally, aCEA-28ζ-sSF6 CAR T cells and aCEA-28ζ-GFP CAR T cells evinced equal proliferation following specific stimulation with CEA^+^ BxPC-3 cells (Figure 2D). In summary, CAR T cells engineered to release soluble SLAMF6 isoform 2 showed augmented IFN-γ secretion upon CAR engagement by target cells.

### 3.3. CAR T Cells Releasing Soluble SLAMF6 Isoform 2 Show Augmented Functional Persistence Under Repetitive Stimulatory Conditions

The functional persistence of SLAMF6-releasing CAR T cells was investigated using the well-established in vitro “stress-test” based on repetitive antigen challenge with CEA^+^ BxPC3 pancreatic cancer cells [21,31,34,35]. During the “stress-test”, CAR T cells expanded, followed by a contraction phase without numerical difference between CAR T cells with and without engineered SLAMF6 (Figure 3A). Importantly, aCEA-28ζ-sSF6 CAR T cells retained their cytotoxic capacity until the third stimulation round, eliminating the cancer cells and significantly outperforming the control aCEA-28ζ-GFP CAR T cells (Figure 3A). The CAR was still expressed on equal levels by both CD4+ and CD8+ cells among aCEA-28ζ-sSF6 CAR T cells and aCEA-28ζ-GFP CAR T cells (Figure 3B and Supplemental Appendix A). In addition, the expression of the cytotoxic effector molecules granzyme B and perforin was similar in CD8^+^ CAR T cells with and without engineered SLAMF6 (Supplemental Appendix A). Remarkably, the expression of bona fide exhaustion-related transcription factors TOX- and IRF-4 was significantly less in aCEA-28ζ-sSF6 CAR T cells relative to control CAR T cells following three rounds of repetitive stimulation with tumor cells (Figure 3C,D). In addition, aCEA-28ζ-sSF6 CAR T cells showed a less differentiated CD62L^+^CD45RO^+^ central memory phenotype upon repetitive stimulation (Figure 3E). Furthermore, the percentage of highly functional PD-1^-^TIGIT^-^ CAR T cells was higher among CAR T cells with engineered SLAMF6 as compared to control CAR T cells after the third round of re-stimulation, whereas the percentage of dysfunctional CD8^+^ PD-1^+^TIGIT^+^ CAR T cells was higher among conventional CAR T cells (Figure 3F and Supplemental Appendix A). By contrast, no differences regarding LAG3 or TIM3 expression were recorded (Supplemental Appendix A). The expression of endogenous SLAMF6 receptor on both CD8^+^ and CD4^+^ CAR T cells with engineered SLAMF6 was higher than on control CAR T cells (Figure 3G and Supplemental Appendix A). This could indicate a positive feedback loop of soluble SLAMF6 release and the expression of the corresponding SLAMF6 receptor. In aggregate, CAR T cells releasing soluble SLAMF6 exhibited augmented functional persistence, an extended killing capacity, and a less exhausted phenotype under repetitive stimulatory conditions compared to conventional CAR T cells.

### 3.4. CAR T Cells Releasing Soluble SLAMF6 Isoform 2 Exhibit Enhanced Functionality Against Melanoma Cells

To explore the generality of our observation, we employed CAR T cells targeting melanoma cells through the well-characterized CSPG4-specific CAR (Figure 4A) [32]. Both aCSPG4-28ζ-sSF6 CAR T cells and aCSPG4-28ζ-GFP CAR T cells showed robust CAR and SLAMF6 expression (Supplemental Appendix A), but only aCSPG4-28ζ-sSF6 CAR T cells evinced soluble SLAMF6 production (Figure 4B). Functionality testing of CSPG4-specifc CAR T cells yielded similar results as obtained with the respective CEA CAR T cells. CD4^+^ T cells engineered with the aCSPG4-28ζ CAR and soluble SLAMF6 displayed a greater magnitude of CD25 upregulation in response to CSPG4^+^ A375M melanoma cells in comparison to the respective CAR T cells without SLAMF6 release; no significant difference was observed for engineered CD8^+^ CAR T cells (Figure 4C). CAR-mediated cytotoxicity and proliferative capacity did not significantly differ between CAR T cells with and without SLAMF6 release (Supplemental Appendix A). IFN-γ secretion in response to CSPG4^+^ A375M melanoma cells was enhanced in CAR T cells with engineered SLAMF6 release compared to control CAR T cells (Figure 4D), while IL-2 release was not affected (Figure 4D). In accordance with the CEA-specific CAR T cells, aCSPG4-28ζ-sSF6 CAR T cells were eminently superior to aCSPG4-28ζ-GFP control CAR T cells in melanoma cell elimination upon several rounds of re-stimulation, while no significant differences in CAR T cell expansion or persistence were observed (Figure 4E). Taken together, the enhanced functionality of CAR T cells with engineered soluble SLAMF6 was also corroborated for melanoma-targeting CAR T cells.

## 4. Discussion

Soluble SLAMF6 has been proven to exert a tangible impact on T cell functionality [29]. Depending on the splice variant, soluble SLAMF6 isoform 2 boosts and isoform 1 inhibits T cell effector functions [30]. Since acquired T cell dysfunctionality (“exhaustion”) prevents successful CAR T cell therapy of solid tumors, we aimed at boosting and extending CAR T cell activation by establishing an auto-stimulatory loop composed of constitutively secreted stimulatory SLAMF6 isoform 2 to augment and maintain CAR T cell activation while preventing exhaustion.

To allow clinical translation, we developed a one-step manufacturing procedure relying on a retroviral vector driving robust expression of both a cancer cell-specific CAR and the soluble SLAMF6 isoform 2. The stimulatory loop seems to be auto-stimulatory as engineered T cells increased the endogenous membrane-bound SLAMF6, serving as a receptor of soluble isoform 2. Also, the loop acted positively on the functional capacities of engineered T cells with respect to the augmented CAR, triggering upregulation of the IL-2 receptor alpha chain CD25 and the enhanced IFN-γ secretion. Notably, SLAMF6 engineered CAR T cells evinced a refined functional persistence outperforming conventional CAR T cells in repetitive killing of cognate cancer cells. Remarkably, engineered release of soluble SLAMF6 produced lower levels of exhaustion-related transcription factors TOX and IRF-4, a major central memory population of CAR T cells, and reduced expression of the exhaustion-related receptors PD-1 and TIGIT. The observation holds true for CAR T cells of two different targeting specificities, adenocarcinoma-associated CEA and melanoma-associated CSPG4, demonstrating the generality of this concept. For both CAR T cell species, co-expression of soluble SLAMF6 isoform 2 improved CAR T cell functionality, particularly during repetitive antigen engagement, and counteracted T cell exhaustion by preventing upregulation of PD-1 and TIGIT, providing a robust circuit for extending CAR T cell functionality under stimulatory conditions.

Several lines of evidence indicate that soluble SLAMF6 isoform 2 can boost T cell functionality. Eisenberg et al. first demonstrated that adding soluble SLAMF6 augmented anti-melanoma CD8^+^ T cell effector performance, produced more IFN-γ, and displayed enhanced cytotoxicity in vitro and in vivo [29]. Moreover, Hajaj et al. identified the truncated splice isoform SLAMF6 Δ17-65 as the crucial mediator of SLAMF6-induced refinement of T cell cytotoxicity [30]. Mechanistically, SLAMF6 has been described as a T cell co-receptor clustering with the T cell receptor to amplify T cell activity upon stimulation through the T cell receptor [36]. The beneficial effect of SLAMF6 is mirrored by the clinical observation that high SLAMF6 expression in the tumor tissues of patients with breast cancer and melanoma is associated with superior progression-free survival and overall survival [37]. Accordingly, high SLAMF6 levels are a surrogate indicator for immune-favorable tumor microenvironments of breast cancer and melanoma; SLAMF6^high^ tumors are significantly enriched with gene expression patterns associated with T cell activation and effector functions [37]. Finally, T cells expressing CARs consisting of the canonical CD3ζ chain and the intracellular signaling part of SLAMF6 exhibited superior functionality in comparison to conventional CAR T cells [38].

Contrary to studies demonstrating beneficial effects of SLAMF6 on T cell functionality, other reports implicate SLAMF6 as a driver of T cell exhaustion, conferring a largely negative impact on T cell functionality in cancer. First, Yigit et al. demonstrated that anti-SLAMF6 treatment improved in vivo T cell activity against chronic lymphocytic leukemia and melanoma by opposing the increase in exhaustion-related receptors, such as PD-1 [39]. In this line, the adoptive transfer of gp100-specific SLAMF6^−/−^ T cells mediated durable melanoma regression, outperforming T cells of the same specificity with preserved SLAMF6 expression [40]. In vitro, gp100-specific SLAMF6^−/−^ T cells evinced a higher cytotoxic capacity and a higher secretion of IFN-γ in response to gp100-expressing melanoma cells [40]. Remarkably, PD-1 expression was increased in 7-day activated gp100-specific SLAMF6^−/−^ T cells compared to T cells with wild-type SLAMF6 expression [40].

In contrast to these studies, we engineered CAR T cells to specifically release the soluble version of the stimulatory SLAMF6 Δ17-65 isoform 2 with the aim of boosting CAR T cell functionality, particularly during repetitive antigen stimulation. Accordingly, SLAMF6 isoform 2 engineered CAR T cells showed a higher proportion of PD-1^-^ TIGIT^-^ CAR T cells. In general, PD-1^-^ TIGIT^-^ T cells display increased functionality, owing to simultaneous resistance to the inhibitory checkpoints PD-1/PD-1L and TIGIT/CD155 [16,41]. An extensive head-to-head comparison of CAR T cells with double knockdown of PD-1 and either LAG3, TIM3, or TIGIT in mice bearing NALM6 leukemia showed superior therapeutic efficacy of CAR T cells with simultaneous knockdown of PD-1 and TIGIT [16]. Importantly, this was not the case for CAR T cells with combined PD-1 and LAG3 or simultaneous PD-1 and TIM3 knockdown [16]. In line with these reports, we recorded augmented functionality and a higher frequency of PD-1^-^TIGIT^-^ CAR T cells with engineered SLAMF6 after several rounds of stimulation.

In general, enhancing the power of CAR T cell products may simultaneously raise the risk of toxicities caused by CAR T cells. In particular, constitutive secretion of sSLAMF6 may pose risks of T cell overactivation due to excessive cytokine release or off-tumor effects. Potential strategies to obviate those concerns prior to clinical translation encompass the use of inducible expression systems, safety switches, or transient delivery platforms. Regarding inducible expression systems, the CAR-signaling induced NFAT-driven payload expression system called TRUCK (T cells redirected for universal cytokine signaling) could be employed to render sSLAMF6 expression inducible by CAR signaling [34]. Incorporating safety switches, such as truncated epidermal growth factor receptor targeted by the monoclonal antibody cetuximab, constitutes another safety mechanism [42]. Finally, transient CAR expression as conferred by mRNA-electroporation poses another avenue to enhance the safety of CAR T cells releasing sSLAMF6 [43].

This study holds an important limitation due to the lack of in vivo testing regarding CAR T cell functionality. Here, we solely focus on procuring seminal engineering and functional data about CAR T cells co-expressing soluble SLAMF6. Apart from transgene expression and functional validation using canonical T cell functionality assays and repetitive antigen stimulation, we provide some mechanistic data on how soluble SLAMF6 mitigates exhaustion in CAR T cells. Case in point, CAR T cells releasing sSLAMF6 exhibited higher percentages of PD-1- and TIGIT-double negative CAR T cells as well as lower levels of exhaustion-related transcription factors IRF-4 and TOX. Suitable animal models meant to be employed in follow-up studies on in vivo testing of CAR T cells co-expressing soluble SLAMF6 would comprise an orthotopic xenograft model of pancreas carcinoma, such as published by Ma et al. [35], in conjunction with a syngeneic mouse model harboring a fully preserved tumor microenvironment, such as the B16 melanoma model [44]. Animal studies are further required to ascertain whether the in vivo *s*SLAMF6 release is sufficient to boost CAR T cell functionality. However, systemic sSLAMF6 levels, which could potentially be measured in the serum, could differ from local sSLAMF6, which would be harder to determine. Moreover, an animal model could be helpful to establish whether systemic sSLAMF6 release could eventuate in potentially harmful autonomous lymphoproliferation.

In order to contextualize the benefit of SLAMF6-engineering against other exhaustion-reducing strategies, e.g., PD-1 KO, IL-15 co-expression, or dominant-negative regulators, the following points should be considered. First, inhibitory surface receptors, such as PD-1, are frequently co-expressed with other checkpoint molecules like LAG3, TIM3, or TIGIT, suggesting a certain degree of redundancy in checkpoint receptor expression. Hence, targeting a single checkpoint receptor could be undermined by the action of other checkpoint molecules. Furthermore, silencing checkpoint receptors in CAR T cells only protects the engineered T cells without adding extra stimulation. In contrast, the release of soluble molecules, such as SLAMF6, does confer extra stimulation to engineered T cells and, potentially, to tumor-specific endogenous T cells. As for IL-15 co-expression, systemic toxicities and uncontrolled lymphoproliferation have posed a serious caveat to this approach [45]. Unlike sSLAMF6, which fosters expression of the corresponding receptor, IL-15 requires the co-delivery of the IL-15 receptor alpha chain to unfold maximum performance enhancement of CAR T cells [46]. Similarly to knockout of checkpoint molecules, the use of dominant–negative receptors, e.g., dominant-negative TGF-beta receptor, confines the protection from checkpoint molecules to the engineered T cells without adding benefit to endogenous T cells. Moreover, targeting single checkpoints may lead to overcompensation by upregulation of various other checkpoint molecules, as exemplified by the clinical evaluation of PSMA-specific CAR T cells rendered TGF-beta-resistant via a dominant-negative TGF-beta receptor [47]. In this study, the expression of myriad inhibitory molecules in the tumor microenvironment following CAR T cell transfer nullified the activity of TGF beta-resistant CAR T cells, corroborating the necessity for diversified approaches. Basically, the co-expression of soluble molecules together with CARs from one vector, as in our study, has been well-established and facilitates swift translation into a GMP-compliant process without the need for additional modifications, e.g., gene editing. Very recently, CAR T cells releasing IL-18 achieved great success in clinical evaluation, underscoring the feasibility and benefit of equipping CAR T cells with soluble mediators [48].

In conclusion, we present seminal data on engineering CAR T cells with an auto-stimulatory loop comprising the release of immunostimulatory SLAMF6 isoform 2 to bolster functional capacities under conditions of prolonged antigen challenge. Mechanistically, a sizable proportion of CAR T cells display the PD-1^-^TIGIT^-^ phenotype and lower levels of exhaustion-related transcription factors TOX and IRF-4 associated with superior and extended functionality. Finally, the one-vector system encoding both soluble SLAMF6 and the CAR expedites efficient GMP-compliant manufacturing of CAR T cells for clinical evaluation.

## Figures and Tables

**Figure 1 cells-14-00901-f001:**
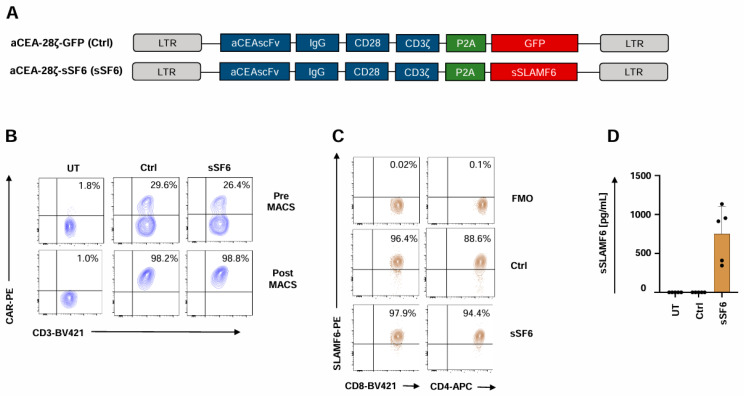
Generation of CAR T cells with release of soluble SLAMF6 isoform 2. (**A**) Schematic outline of CAR constructs. (**B**) CAR expression on CAR T cells (CEA-28ζ-sSF6 = sSF6; CEA-28ζ-GFP = Ctrl) was determined prior (upper panels) and after (lower panels) magnetic cell separation (MACS). Untransduced T cells (UT) served as controls. One representative donor out of six is shown. (**C**) Baseline SLAMF6 expression on CD8^+^ and CD4^+^ CAR T cells (CEA-28ζ-sSF6 = sSF6; CEA-28ζ-GFP = Ctrl) at the start of in vitro assays. FMO (Fluorescence Minus One) controls were used. One representative donor out of four is depicted. (**D**) ELISA-based quantification of soluble SLAMF6 in the supernatants after a 48 h culture of engineered 2 × 10^5^ CAR T cells (CEA-28ζ-sSF6 = sSF6; CEA-28ζ-GFP = Ctrl) in 200 µL medium. Untransduced T cells (UT) served as controls. Data represent means ± SD of five T-cell donors.

**Figure 2 cells-14-00901-f002:**
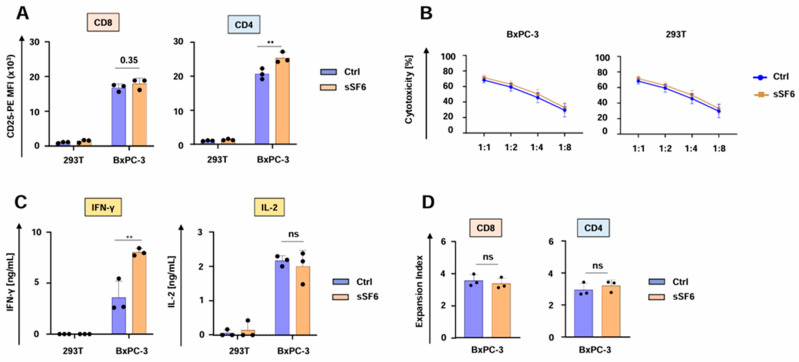
CAR T cells releasing soluble SLAMF6 display enhanced IFNγ secretion. (**A**) Antigen-specific CD25 upregulation in CD8+ and CD4+ CAR T cells (CEA-28ζ-sSF6 = sSF6; CEA-28ζ-GFP = Ctrl) 24 h after co-incubation with BxPC-3 cells and 293 T cells. (**B**) Cytotoxicity of CAR T cells (CEA-28ζ-sSF6 = sSF6; CEA-28ζ-GFP = Ctrl) after a 24 h co-culture with BxPC-3 cells and 293T cells at the indicated effector to target cell ratios. (**C**) CAR-triggered production of IFN-γ (detection limit 50 ng/mL) and IL-2 (detection limit 10 ng/mL) by CAR T cells (CEA-28ζ-sSF6 = sSF6; CEA-28ζ-GFP = Ctrl) stimulated with BxPC-3 cells or 293T cells after 48 h, as measured by ELISA. (**A**–**C**) Data represent means ± SD of three donors, *p* values were calculated by two-way ANOVA, ** indicates *p* ≤ 0.01, and ns indicates not significant. (**D**) Expansion index of CD8^+^ and CD4^+^ CAR T cells (CEA-28ζ-sSF6 = sSF6; CEA-28ζ-GFP = Ctrl) labeled with the Cell Proliferation Dye eFluor^®^ 450 and activated with BxPC-3 cells for five days. Expansion index was calculated based on dye dilution). Data represent means ± SD of three donors, *p* values were calculated by Student’s t-test in conjunction with Welch’s correction, and ns indicates not significant.

**Figure 3 cells-14-00901-f003:**
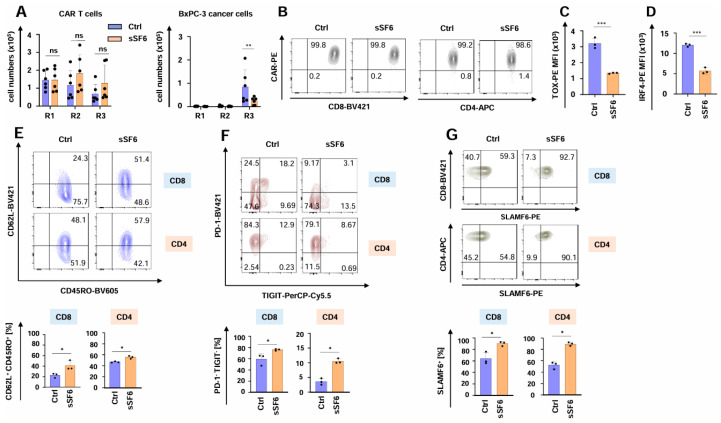
Release of soluble SLAMF6 improves the functional persistence of CAR T cells. (**A**) CAR T cells (CEA-28ζ-sSF6 = sSF6; CEA-28ζ-GFP = Ctrl) (1 × 10^5^ CAR T cells) underwent three rounds (R1-R3) of stimulation with GFP-labeled CEA^+^ BxPC-3 cells (1 × 10^5^ tumor cells at the beginning of each round). At the end of each round, CAR T cells (live CD3^+^ CAR^+^ cells) (left panel) and BxPC-3 cells (right panel) were quantified by flow cytometry using counting beads. Data represent means ± SD of six donors, *p* values were calculated by two-way ANOVA, ns indicates not significant, * *p* ≤ 0.05, and ** *p* ≤ 0.01. (**B**–**E**) Phenotypic analysis of CAR T cells (CEA-28ζ-sSF6 = sSF6; CEA-28ζ-GFP = Ctrl) undergoing three rounds of antigen-stimulation with unlabeled BxPC-3 cells. (**B**) At the end of round three, CAR T cells were stained for CAR expression. One representative donor out of three is shown. (**C**,**D**) At the end of round three, CAR T cells were stained for TOX (**C**) and IRF-4 (**D**). Data represent means ± SD of three donors, *p* values were calculated by Student’s *t*-test, *** indicates *p* ≤ 0.001. (**E**) Effector memory cell differentiation of CAR T cells as determined by CD62L and CD45RO staining at the end of stimulation round three. The bar graph below shows the percentages of CM = central memory (CD45RO^+^ CD62L^+^) CD8^+^ and CD4^+^ CAR T cells. (**F**) At the end of round three, CAR T cells were stained for PD-1 and TIGIT expression. The bar graph below shows the percentages of PD-1/TIGIT double negative (PD-1^-^ TIGIT^-^) CD8^+^ and CD4^+^ CAR T cells. (**G**) At the end of round three, CAR T cells were stained for SLAMF6 in CD8^+^ (**upper panels**) and CD4^+^ (**lower panels**) CAR T cells. The bar graph below shows the percentages of SLAMF6^+^ CD8^+^ and CD4^+^ CAR T cells. (**C**–**E**) One representative donor out of three is shown. Data represent means ± SD of three donors, *p* values were calculated by Student’s *t*-test in conjunction with Welch’s correction, ns indicates not significant, * *p* ≤ 0.05, and *** *p* ≤ 0.001.

**Figure 4 cells-14-00901-f004:**
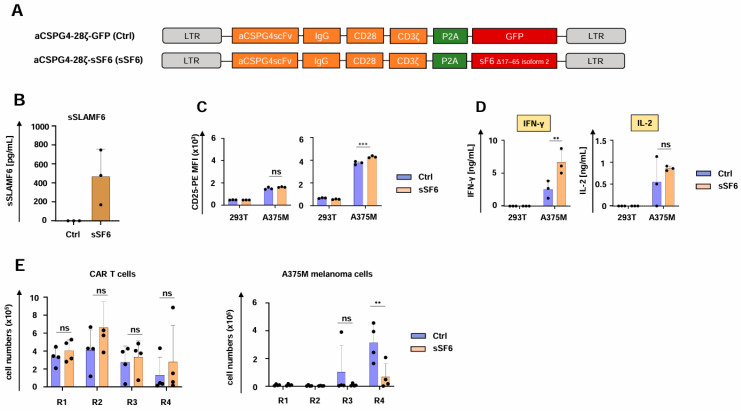
CAR T cells releasing soluble SLAMF6 exhibit enhanced functionality against melanoma. (**A**) Schematic of CAR constructs. (**B**) ELISA-based quantification of soluble SLAMF6 in the supernatants after a 48 h culture of engineered CAR T-cells (CSPG4-28ζ-sSF6 = sSF6; CSPG4-28ζ-GFP = Ctrl). Data represent means ± SD of three T-cell donors. (**C**) Antigen-specific CD25 upregulation in CD8^+^ (**left panel**) and CD4^+^ CAR T cells (**right panel**) (CSPG4-28ζ-sSF6 = sSF6; CSPG4-28ζ-GFP = Ctrl) 24 h after co-incubation with A375M cells and 293 T cells. (**D**) CAR-triggered production of IFN-γ and IL-2 by CAR T cells (CSPG4-28ζ-sSF6 = sSF6; CSPG4-28ζ-GFP = Ctrl) stimulated with A375M cells or 293T cells after 48 h, as measured by ELISA. (**E**) CAR T cells (CSPG4-28ζ-sSF6 = sSF6; CSPG4-28ζ-GFP = Ctrl) (1 × 10^5^ CAR T cells) underwent four rounds (R1-R4) of stimulation with GFP-labeled CSPG4^+^ A375M melanoma cells (1 × 10^5^ tumor cells at the beginning of each round). At the end of each round, CAR T cells (live CD3^+^ CAR^+^ cells) (**left panel**) and A375M cells (**right panel**) were quantified by flow cytometry with counting beads (absolute count). (**B**–**E**) Data represent means ± SD of at least three donors, *p* values were calculated by two-way ANOVA, ns indicates not significant, ** *p* ≤ 0.01, and *** *p* ≤ 0.001.

## Data Availability

The original contributions presented in this study are included in the article/Appendix A. Further inquiries can be directed to the corresponding author.

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
