# Peer review of "Chimeric Antigen Receptor (CAR) T Cells Releasing Soluble SLAMF6 Isoform 2 Gain Superior Anti-Cancer Cell Functionality in an Auto-Stimulatory Fashion"

_cells, 2025, doi:10.3390/cells14120901_

Round 1
Reviewer 1 Report
Comments and Suggestions for Authors
The manuscript, "Chimeric antigen receptor (CAR) T cells releasing soluble SLAMF6 isoform 2 gain superior anti-cancer cell functionality in an auto-stimulatory fashion" by Harrer et al. describes a novel CAR T cell armoring strategy capable of producing a moderate increase in central memory phenotype cells and IFNγ production upon CAR stimulation. More importantly, armoring with soluble SLAMF6 increased functional persistence of chronically stimulated CAR T cells accompanied by an increase in PD1-, TIGIT- CAR T cells, indicating that soluble SLAMF6 may reduce CAR T cell exhaustion. These data are consistent with prior reports showing that a soluble isoform of SLAMF6 (Δ17-65 isoform 2) that can be naturally produced through alternative splicing can sustain CD8 T cell function. Overall, this armoring strategy is novel and promising, although not terribly innovative given the known effects of soluble SLAMF6 on T cells. Experiments presented are well controlled, results are adequately described, and statistical methods are appropriate. My only significant critique of this manuscript is that it lacks in vivo testing of CAR T cell function. This is particularly important because, as the authors appropriately discuss, there is evidence that SLAMF6 can limit anti-tumor T cell activity in vivo and also because the authors introduce the manuscript by invoking the need for better CAR T cell therapies for solid tumors. Additionally, more evidence of CAR T cell exhaustion following repeated stimulation would strengthen the manuscript... perhaps intracellular staining for IFNγ or other cytokines. Two more very minor points to consider when revising the manuscript are:
- The table in Figure 1A is confusing as it led me to expect that all 3 isoforms would be expressed to compare effects of each. I don't think this table is necessary since the isoforms are well described in the text.
- It would probably be better to quantify PD1+, TIGIT+ cells in Figure 3D instead of PD1-,TGIT- cells.
Author Response
Point to point response:
First, we would like to thank all reviewers for reading our manuscript.
Reviewer 1:
My only significant critique of this manuscript is that it lacks in vivo testing of CAR T cell function. This is particularly important because, as the authors appropriately discuss, there is evidence that SLAMF6 can limit anti-tumor T cell activity in vivo and also because the authors introduce the manuscript by invoking the need for better CAR T cell therapies for solid tumors.
We absolutely agree with the reviewer on the fact that in vivo testing of CAR T cells co-expressing soluble SLAMF6 is crucial. With our manuscript, we provide an appropriate base for future animal experiments. Nevertheless, we will address the in vivo testing in a follow-up study. We added a paragraph to the discussion describing the lack of animal work as a limitation of this work. However, we think that this study contains enough novel aspects to be considered for publication without in vivo experiments.
Additionally, more evidence of CAR T cell exhaustion following repeated stimulation would strengthen the manuscript... perhaps intracellular staining for IFNγ or other cytokines.
We agree with the reviewer that more evidence of CAR T cell exhaustion following repeated stimulation would strengthen the manuscript. Phenotypically, we found a decreased upregulation of exhaustion related molecules PD-1 and TIGIT on SLAMF 6 secreting CAR T cells relative to conventional CAR T cells. To gain more insight into exhaustion we stained for bona fide exhaustion-related transcription factors TOX and IRF4. After three rounds of repetitive stimulation, SLAMF6 secreting CAR T cells expressed significantly expressed less TOX and IRF4 corroborating the less exhausted phenotype as compared to conventional CAR T cells (Figure 3C and D). We know from previous experience that cytokine secretion (even by intracellular staining) is extinguished in the last round of stress testing with killing capacity being the only preserved T cell function.
The table in Figure 1A is confusing as it led me to expect that all 3 isoforms would be expressed to compare effects of each. I don't think this table is necessary since the isoforms are well described in the text.
We apologize for the confusing representation of the various isoforms, and we accordingly removed the table from Figure 1 A.
It would probably be better to quantify PD1+, TIGIT+ cells in Figure 3D instead of PD1-,TGIT- cells.
We agree with the reviewer that it would be good to quantify PD1+, TIGIT+ cells. Indeed, in the CD8-positve compartment, PD1+, TIGIT+ cells were significantly more prevalent in the conventional CAR T cell cohort. Due to donor variability the percentage of PD1+, TIGIT+ cells clearly trended towards less prevalence in the SLAMF6 CAR T cell group, but did not meet significance due to donor variability. Hence, we put PD1+, TIGIT+ cells in supplementary Figure 3C.
Reviewer 2 Report
Comments and Suggestions for Authors
This manuscript presents a novel and promising approach to improve the persistence and functionality of CAR T cells against solid tumors by engineering them to secrete the Δ17–65 isoform of soluble SLAMF6. The authors demonstrate that this strategy enhances IFN-γ secretion, promotes a central memory phenotype, reduces PD-1/TIGIT exhaustion markers, and improves tumor clearance after repetitive stimulation.
The concept is timely and addresses a significant limitation in the field: the limited efficacy of CAR T cells in solid tumor settings due to exhaustion and poor persistence. The comprehensive data include two different CAR constructs targeting distinct tumor antigens, adding robustness. However, to meet publication standards and enhance reproducibility and clarity, the authors must address several concerns related to figure annotation, statistical interpretation, and biological variability, as outlined below.
Major Comments:
1. The study’s findings are based entirely on in vitro co-culture assays using human primary T cells, which provides valuable mechanistic insight. However, the absence of validation in more physiologically relevant systems limits the translational depth of the study. To strengthen the clinical relevance and support potential therapeutic application, the authors are encouraged to validate their findings using: In vivo xenograft tumor models, which would allow assessment of CAR T cell persistence, trafficking, and tumor control in a systemic context, or Patient-derived tumor organoids or 3D co-culture platforms, which can better mimic the complex tumor microenvironment while retaining human-specific interactions.
2. While phenotypic improvements are clearly documented, mechanistic dissection is limited. The authors should consider:
Incorporating transcriptional profiling, or
Evaluating key signaling pathways (e.g., STAT, mTOR, NFAT) to define how SLAMF6 modulates T cell memory and exhaustion.
3. Constitutive secretion of sSLAMF6 may pose risks of T cell overactivation, cytokine release, or off-tumor effects.
The authors should include a brief discussion on safety, such as inducible expression systems, safety switches, or transient delivery platforms for clinical translation.
4. The authors consistently use SEM with small sample sizes (n = 3), which underrepresents biological variability. In several cases, large SEM bars obscure potential trends. Additionally, there is no indication that the authors tested for normality or equal variance, nor did they mention using Welch’s correction or non-parametric alternatives, even when variance clearly differed between groups.
Minor Comments and Suggestions:
1. A side-by-side comparison with other exhaustion-reducing strategies (e.g., PD-1 KO, IL-15 co-expression, or dominant-negative regulators) would contextualize the benefit of SLAMF6 engineering.
2. Endogenous SLAMF6 expression increases upon SLAMF6 isoform 2 secretion. This suggests a potential autocrine feedback loop that could reinforce signaling. A short mechanistic discussion would enhance reader understanding.
3. The dual-expression single-vector design is a technical strength. The authors may briefly comment on its GMP compatibility, viral titers, and clinical scalability.
4. A schematic illustrating the SLAMF6 auto-stimulatory loop and its effect on memory and exhaustion could improve accessibility for readers and strengthen the conclusion section.
Figure-Specific Comments
The data are compelling, but several visual and statistical presentation issues limit interpretability:
Figure 1 (Panels B–D)
Missing ordinate labels (e.g., CAR, CD4, SLAMF6) compromise clarity.
FMO/isotype controls for SLAMF6 should be included to support gating.
Ctrl bar in Panel D is at baseline; consider adding a mock-transduced or no-vector control to confirm specificity.
The red overlay in Panel C is too intense—use a lighter shade to improve visualization.
Figure 2
Panel A: Clarify "CD25 MFI (×10³)", and add exact p-values.
Panel B: Cytotoxicity curves are well matched; legend symbols should match line styles.
Panel C: IFN-γ increase is clear; include ELISA limits and replicate variation.
Panel D: Define how the expansion index was calculated (e.g., based on dye dilution).
Global Note: Use SD instead of SEM, and overlay individual donor dots to show biological variability (n = 3). Also, the red color is visually dominant—suggest a lighter tone.
Figure 3
Panel A: CAR T cell numbers do not significantly differ; however, variance is high in the sSF6 group. Plot individual donors or use SD to avoid masking trends.
Panels B–E:
Missing ordinate labels (CAR, PD-1, TIGIT, CD62L, CD45RO).
Panel D: Statistical methods should be clarified—if assumptions are violated, Welch’s t-test or Mann–Whitney should be considered.
Panel E: Extremely high SLAMF6 positivity (>90%) warrants explanation—distinguish surface-bound from total protein, and confirm gating strategy.
Figure 4
Panel A: Nicely presented; suggest labeling isoform Δ17–65 explicitly.
Panel B: Display donor-level spread.
Panel C: CD25 is upregulated in CD4⁺ cells; simplify panel by removing boxed text (merge into legend).
Panel D: High variability in IL-2 responses again underscores the need for dot plots.
Panel E: Tumor clearance enhancement is observed at R4, but CAR T cell persistence shows high SEM. Clarify whether this is a normalized or an absolute count.
Author Response
Point to point response:
First, we would like to thank all reviewers for reading our manuscript.
Reviewer 2:
The study’s findings are based entirely on in vitro co-culture assays using human primary T cells, which provides valuable mechanistic insight. However, the absence of validation in more physiologically relevant systems limits the translational depth of the study. To strengthen the clinical relevance and support potential therapeutic application, the authors are encouraged to validate their findings using: In vivo xenograft tumor models, which would allow assessment of CAR T cell persistence, trafficking, and tumor control in a systemic context, or Patient-derived tumor organoids or 3D co-culture platforms, which can better mimic the complex tumor microenvironment while retaining human-specific interactions.
We absolutely agree with the reviewer on the fact that in vivo testing of CAR T cells co-expressing soluble SLAMF6 is crucial. With our manuscript, we provide an appropriate base for future animal experiments. Nevertheless, we will address the in vivo testing in a follow-up study. We added a paragraph to the discussion describing the lack of animal work as a limitation of this work. However, we think that this study contains enough novel aspects to be considered for publication without in vivo experiments.
While phenotypic improvements are clearly documented, mechanistic dissection is limited. The authors should consider:Incorporating transcriptional profiling, or Evaluating key signaling pathways (e.g., STAT, mTOR, NFAT) to define how SLAMF6 modulates T cell memory and exhaustion:
We agree with the reviewer that more evidence of CAR T cell exhaustion following repeated stimulation would improve the manuscript. Mechanistically, we found a decreased upregulation of exhaustion related molecules PD-1 and TIGIT on SLAMF 6 secreting CAR T cells relative to conventional CAR T cells. To gain more insight into exhaustion we stained for bona fide exhaustion-related transcription factors TOX and IRF4. After three rounds of repetitive stimulation, SLAMF6 secreting CAR T cells expressed significantly expressed less TOX and IRF4 corroborating the less exhausted phenotype as compared to conventional CAR T cells (Figure 3C and D).
Constitutive secretion of sSLAMF6 may pose risks of T cell overactivation, cytokine release, or off-tumor effects. The authors should include a brief discussion on safety, such as inducible expression systems, safety switches, or transient delivery platforms for clinical translation.
We thank the reviewer for raising important points to consider with regard to clinical translation. We added the suggestions of the reviewer into the discussion section (ll 399-410).
- The authors consistently use SEM with small sample sizes (n = 3), which underrepresents biological variability. In several cases, large SEM bars obscure potential trends. Additionally, there is no indication that the authors tested for normality or equal variance, nor did they mention using Welch’s correction or non-parametric alternatives, even when variance clearly differed between groups.
We apologize for the confusion with the statistics and agree with the necessity to clarify. In all main figures SEM was replaced with SD and individual donors were plotted. T tests where always conduced using Welch’s correction. We added this piece of information to the figure legends and methods.
- A side-by-side comparison with other exhaustion-reducing strategies (e.g., PD-1 KO, IL-15 co-expression, or dominant-negative regulators) would contextualize the benefit of SLAMF6 engineering.
We added a corresponding paragraph to the discussion section (ll 426-447).
Endogenous SLAMF6 expression increases upon SLAMF6 isoform 2 secretion. This suggests a potential autocrine feedback loop that could reinforce signaling. A short mechanistic discussion would enhance reader understanding.
We added Figure 5 to illustrate this potential autocrine loop.
The dual-expression single-vector design is a technical strength. The authors may briefly comment on its GMP compatibility, viral titers, and clinical scalability.
We incorporating this interesting thought of the reviewer into the discussion (ll 448-452).
A schematic illustrating the SLAMF6 auto-stimulatory loop and its effect on memory and exhaustion could improve accessibility for readers and strengthen the conclusion section.
We added a corresponding schematic Figure 5 to the discussion section.
Figure-Specific Comments:
We apologize for the issues regarding visual and statistical presentation. We thank the reviewer for the valuable suggestions, which were all implemented in the revised manuscript.
- We changed the color of the bars to a lighter tone.
- We replaced SEM by SD and overlayed individual donor dots to show biological variability
Figure 1:
- We added missing ordinate labels to panel B-D.
- FMO control for SLAMF6 now included.
- Ctrl bar in Panel D is at baseline due to lacking SLAMF6 secretion. Untransduced T cells as additional controls now shown
Figure 2:
- CD25-PE MFI (×10³) calrified, and add exact p-values added.
- confirmed that legend symbols should match line styles
- ELISA limits and replicate variation included
- calculation of expression index now defined in the figure legend
Figure 3:
- Individual donors and SD shown in panel A
- Missing ordinate labels added to panels B-E
- In panel D use of t Test with Welch’s correction clarified in the legend
- Positive feedback-loop assumed, if surface-bound protein played a significant role, the same phenomenon would be observed in Figure 1C (here equal levels); gating strategies added to supplementary Figure 1E
Figure 4:
- Isoform Δ17–65 explicitly labeled in panel A
- Donor-level spread displayed in panel B
- boxed text merged into legend in panel C
- Individual donors and SD shown in panel D
- absolute count in panel E clarified in the leged
Reviewer 3 Report
Comments and Suggestions for Authors
In the manuscript titled "Chimeric antigen receptor (CAR) T cells releasing soluble SLAMF6 isoform 2 gain superior anti-cancer cell functionality in an auto-stimulatory fashion” by Harrer et. al. the authors aim to investigate whether engineering CAR T cells to secrete a modified, immunostimulatory form of soluble SLAMF6 can enhance their anti-tumor function in solid cancers. Using CD28-based CAR T cells, the authors observed an increased cytokine secretion and activation markers when targeting pancreatic and melanoma cells. Under conditions of repetitive antigen encounter, these engineered cells showed improved long-term killing capacity. The authors further showed that the engineered CAR T cells exhibited a favorable memory phenotype and reduced exhaustion markers. The study proposes a novel auto-stimulatory loop to bolster CAR T cell efficacy in solid tumors. In summary, this is an interesting research work. However, there are minor major concerns that need to be addressed before the manuscript could be considered for the publication. 1) Correction of spelling in Line 112.
2) The font size of the figure labels is too small to be easily readable. 3) It's not very clear why authors choose Isoform 2? 4) Fig. 1, the controls lack non-transduced T cells for baseline. 5) In the cytotoxicity assay, did the authors study the results beyond 24 hours particularly to study the persistence and more importantly in solid tumor models where killing may be slower. 6) Why didn't the authors use CD69 which in combination with CD25 would have given a more comprehensive view of CAR T activation dynamics. 7) The discussion looks very similar to the Introduction, and doesn't discuss the results provided. 8) The work is promising however lack of in vivo data limits the ability to assess the therapeutic efficacy, persistence, and safety of SLAMF6-secreting CAR T cells within a physiologically relevant tumor microenvironment.
Author Response
Point to point response:
First, we would like to thank all reviewers for reading our manuscript.
Reviewer 3:
1) Correction of spelling in Line 112.
Spelling error corrected.
2) The font size of the figure labels is too small to be easily readable.
We increased the size of figure labels.
3) It's not very clear why authors choose Isoform 2?
We wanto to draw the attention of the reviewer to the following paragraph at the end of the introduction (ll 71-76):
Three splice isoforms of SLAMF6 with different ectodomains have been described in T cells, with isoform 1 being the predominant variant expressed at the highest level, where-as isoform 2 lacks amino acids 17–65 (Δ17–65) of the variable region, and isoform 3 is devoid of the entire variable chain (ΔExon 2) [30]. Importantly, the SLAMF6 splice variants display different activities. The canonical SLAMF6 promotes T cell exhaustion while the Δ17–65 isoform 2 sustains T cell functionality [30].
4) Fig. 1, the controls lack non-transduced T cells for baseline.
Untransduced T cells were added as baseline.
5) In the cytotoxicity assay, did the authors study the results beyond 24 hours particularly to study the persistence and more importantly in solid tumor models where killing may be slower.
To model killing at later timepoints, which is indeed more important in the realm of solid tumor, we designed our “stress-test” (Figure 3A and Figure 4E) to track killing capacity for up to three round of repetitive stimulation with BxPC-3 pancreatic cancer cells.
6) Why didn't the authors use CD69 which in combination with CD25 would have given a more comprehensive view of CAR T activation dynamics.
CD69 is a marker of early T cell activation. Co-expression of soluble SLAMF6 is intended to ameliorate T cell exhaustion occurring after several days of antigen encounter. Thus, we did not focus on upregulation of CD69.
7) The discussion looks very similar to the Introduction, and doesn't discuss the results provided.
We added several new aspects to the discussion section.
8) The work is promising however lack of in vivo data limits the ability to assess the therapeutic efficacy, persistence, and safety of SLAMF6-secreting CAR T cells within a physiologically relevant tumor microenvironment.
We absolutely agree with the reviewer on the fact that in vivo testing of CAR T cells co-expressing soluble SLAMF6 is crucial. With our manuscript, we provide an appropriate base for future animal experiments. Nevertheless, we will address the in vivo testing in a follow-up study. We added a paragraph to the discussion describing the lack of animal work as a limitation of this work. However, we think that this study contains enough novel aspects to be considered for publication without in vivo experiments.
Round 2
Reviewer 1 Report
Comments and Suggestions for Authors
Revisions discussing future in vivo experiments should further address potential for differences between in vitro results and potential in vivo results based on prior studies.
Figures are low resolution and must be improved prior to publication.
Author Response
Revisions discussing future in vivo experiments should further address potential for differences between in vitro results and potential in vivo results based on prior studies.
We extended the discussion section accordingly (ll. 426 to 432).
Figures are low resolution and must be improved prior to publication.
We improved the figure resolution.
Reviewer 2 Report
Comments and Suggestions for Authors
The authors have thoroughly addressed all reviewer comments thoughtfully and comprehensively. The revised manuscript shows improved clarity, enhanced figure quality, and strengthened mechanistic data supporting the role of soluble SLAMF6 in mitigating CAR T cell exhaustion. While in vivo validation remains a limitation, the study provides a solid mechanistic foundation and is appropriate for publication in its current form.
Comments on the Quality of English LanguageThe English language is generally clear and facilitates understanding, although minor polishing could improve readability. Clear sentence structure in most sections.
Author Response
We thank the reviewer for approving our manuscript.
Reviewer 3 Report
Comments and Suggestions for Authors
The authors have addressed all my concerns and have made suggested changes in the manuscript. I, therefore, recommend the manuscript suitable for publication in its revised form.
Author Response

(The authors gave the same response as above.)
